# Antibacterial Properties of Nanoparticles in Dental Restorative Materials. A Systematic Review and Meta-Analysis

**DOI:** 10.3390/medicina56020055

**Published:** 2020-01-29

**Authors:** Elena Ferrando-Magraner, Carlos Bellot-Arcís, Vanessa Paredes-Gallardo, José Manuel Almerich-Silla, Verónica García-Sanz, Mercedes Fernández-Alonso, José María Montiel-Company

**Affiliations:** 1Orthodontics Teaching Unit, Department of Stomatology, Faculty of Medicine and Dentistry, University of Valencia, 46010 Valencia, Spain; elenaferrandomagraner@gmail.com (E.F.-M.); carlos.bellot@uv.es (C.B.-A.); veronica.garcia-sanz@uv.es (V.G.-S.); 2Preventive Dentistry Teaching Unit, Department of Stomatology, Faculty of Medicine and Dentistry, University of Valencia, 46010 Valencia, Spain; jose.m.almerich@uv.es (J.M.A.-S.); jose.maria.montiel@uv.es (J.M.M.-C.); 3GROC·UJI, Institute of New Imaging Technologies, Universitat Jaume I, 12071 Castellón, Spain; fernande@uji.es

**Keywords:** nanoparticles, nanotechnology, orthodontic adhesive, dental bonding materials, antibacterial

## Abstract

*Background and Objectives:* Nanotechnology has become a significant area of research focused mainly on increasing the antibacterial and mechanical properties of dental materials. The aim of the present systematic review and meta-analysis was to examine and quantitatively analyze the current evidence for the addition of different nanoparticles into dental restorative materials, to determine whether their incorporation increases the antibacterial/antimicrobial properties of the materials. *Materials and Methods:* A literature search was performed in the Pubmed, Scopus, and Embase databases, up to December 2018, following PRISMA (Preferred Reporting Items for Systematic Review and Meta-Analysis) guidelines for systematic reviews and meta-analyses. *Results:* A total of 624 papers were identified in the initial search. After screening the texts and applying inclusion criteria, only 11 of these were selected for quantitative analysis. The incorporation of nanoparticles led to a significant increase (p-value < 0.01) in the antibacterial capacity of all the dental materials synthesized in comparison with control materials. *Conclusions:* The incorporation of nanoparticles into dental restorative materials was a favorable option; the antibacterial activity of nanoparticle-modified dental materials was significantly higher compared with the original unmodified materials, TiO_2_ nanoparticles providing the greatest benefits. However, the high heterogeneity among the articles reviewed points to the need for further research and the application of standardized research protocols.

## 1. Introduction

Nowadays, restorative dentistry is mainly based on adhesive dentistry, in which resin-based materials are the first choice for restoration procedures [1].

However, this type of material tends to accumulate more biofilm than other restoration materials and dental hard tissues, such as enamel [2,3,4].

The use of dental restorative materials suffers an inherent problem that develops over time, nearly half of all dental restorations fail within 10 years, and replacing them accounts for 50–70% of all restorative dentistry [1,4]. The colonization by cariopathogenic bacteria at the restoration/tooth interface and the consequent acid attack produces secondary caries, reducing the durability of the restoration [5,6].

Similar problems occur with orthodontic applications; the placement of fixed orthodontic apparatus facilitates the accumulation and proliferation of cariogenic bacteria [1,7], which produce organic acid and cause enamel demineralization, manifesting as white spot lesions (WSL), the first sign of demineralization [8,9,10].

WSL constitutes a major complication in patients undergoing orthodontic treatment with fixed apparatus [11], being its prevalence over 50% [12].

Although non-cavitated lesions are reversible most of the time, they could be potentially irreversible if not diagnosed early or when there is a lack of lesion monitoring [13]. In those cases, preventative measures that do not require patient compliance may be more effective [11].

Given that bacteria are mainly responsible for shortening restorations’ functional life and the appearance of WSL [1], an effective method for preventing enamel demineralization and the appearance of cavitated lesions is the use of dental materials that are resistant to bacterial accumulation and ideally, materials that suppress bacterial activity at the tooth-restoration interface [1,14].

Numerous attempts have been made to develop dental restorative materials that offer good long-term behavior and antibacterial activity without sacrificing their mechanical properties [14,15].

The most important advance in the field of dental materials has been the introduction of nanotechnology [16], which represents a promising area of research in dentistry, mainly aimed at improving dental materials’ antibacterial and mechanical properties [17,18].

Numerous studies have focused on evaluating the antibacterial properties of various nanoparticles and investigating the possibility of incorporating these nanoparticles into dental restorative materials in order to produce restoration materials that prevent bacterial accumulation and so the appearance of secondary caries [12,16,19].

Although a great deal of research has been produced on this subject, it is subject to great variability in terms of the diversity of nanoparticle types assayed and the methodologies applied. To date, no systematic review has been conducted of published research into the properties of nanoparticle-modified dental restorative materials.

Given the increasing interest in nanotechnology in the field of adhesive dentistry, the aim of the present systematic review and meta-analysis was to examine current evidence for the incorporation of different nanoparticles into dental restorative materials in order to enhance their antibacterial properties.

## 2. Material and Methods

This systematic literature review was conducted in accordance with PRISMA (Preferred Reporting Items for Systematic Reviews and Meta-Analyses) guidelines [20].

The research question was: Does the incorporation of nanoparticles into dental restorative materials increase their antibacterial/antimicrobial properties?

### 2.1. Inclusion Criteria

Studies evaluating the antibacterial/antimicrobial properties of nanoparticle-modified dental restorative materials.In vitro studies.

#### For Quantitative Analysis:

Only studies that presented exact data, both at baseline and the end of the study period, in direct contact tests and colony-forming unit (CFU) counts were analyzed.Studies with sample incubation periods of between 0 and 72 h were included.

### 2.2. Exclusion Criteria

Literature reviews, systematic reviews, case series, and editorials.Studies without a suitable control group that would allow evaluation of the effect of incorporating nanoparticles on the antibacterial capacity of the material.Studies that evaluated the antibacterial/antimicrobial capacity of the nanoparticle material by themselves, independently of their incorporation into a dental material.Studies focused on materials used for endodontic sealing, acrylic resins not used in dental adhesive treatments, materials for coating implants, or materials used for provisional restorations.Studies focused on the cement used for dental treatments not involving adhesion (Portland cement, mineral trioxide aggregate (MTA), calcium hydroxide cement).

### 2.3. Search Strategy

An electronic search was conducted in Pubmed, Embase, and Scopus online databases. An electronic search for “grey literature” was also performed in the New York Academy of Medicine Gray Literature Report.

The search terms applied were as follows: (dental OR dentistry OR orthodontic*) AND (adhesive* OR cement* OR composite* OR resin*) AND (nanoparticle* OR nanotechnology) AND (antibact* OR anti-bact* OR antimicro* OR anti-micro* OR antibiofilm* OR anti-biofilm* OR antiinfect* OR anti-infect* OR bactericidal* OR bacteriostatic*).

The electronic search was complemented with a manual search among the reference lists of all the articles identified, in order to locate studies not detected during the initial search. No restrictions were applied regarding the publication year or the language. The search covered all works published up to December 2018. Endnote X7 software (Thompson Reuters, Philadelphia, PA, USA) was used to remove duplicates.

Two qualified researchers (E.F.-M. and C.B.-A.) independently assessed the titles and abstracts of all the articles identified in the initial search. In the event of any doubt, a third (V.G.-S.) reviewer was consulted, and agreement reached. When the information provided by the abstract was insufficient to reach a conclusion, the full article was read before taking a final decision. Afterward, the full texts of the remaining articles were examined by two researchers (E.F.-M. and C.B.-A.) independently, and the reasons for rejecting the studies were recorded.

### 2.4. Data Extraction

In each article reviewed, the following variables were registered: author, publication year, study type, study groups, sample size, nanoparticle type, bonding material assessed, storage conditions, antibacterial tests used, conclusions, and the quality of the study.

### 2.5. Quality Assessment

The methodological quality of each study was assessed by two reviewers (V.P.-G.–J.M.M.-C.) independently, following a protocol adapted from an in vitro systematic review conducted by Sarkis-Onofre et al. (2014) [21]. Quality assessment was based on the presence and adequacy of the following items: sample size calculation, adequate control group, use of materials according to the manufacturer’s instructions, standardized sample production process, standardized antibacterial assessment, evaluation of antibacterial properties by a single operator, and adequate statistical analysis (mean, standard deviation, and p-values present).

If an item was reported in the article, the study was marked with a ‘Y’ (yes) for that specific parameter; if the item was not stated, the study was marked with an ‘N’ (no). Studies with one to three items reported were classified as having a high risk of bias, four or five items as medium risk of bias, and six or seven items as low risk of bias.

### 2.6. Statistical Analysis

For the meta-analysis, a random-effects model was created, and the effect size was calculated using Hedge’s G standardized mean difference. Data were interpreted by ‘rule of thumb’, which took values below 0.5 as a small effect, values of 0.5–0.8 as a medium effect, and values over 0.8 as a large effect.

Heterogeneity was evaluated with the Q-test and I-square value. It was considered that heterogeneity was established when the Q-test generated a p-value below 0.1, and when the I-square value was over 50%. Funnel plots and the classic fail-safe number were used to assess publication bias.

Qualitative analysis was performed with Comprehensive Meta-Analysis V.3 software (Biostat, Inc. Englewood, CO, USA).

To make a precise evaluation of the influence of incorporating nanoparticles into the structure of a dental restorative material on their antibacterial capacity, three different meta-analyses were performed: (1) variations in the antibacterial capacity of orthodontic adhesives due to the incorporation of nanoparticles into their composition; (2) variations in the antibacterial capacity of resin-based dental bonding materials due to the incorporation of nanoparticles into their composition; (3) variations in the antibacterial capacity of all dental restorative materials due to the incorporation of nanoparticles into their composition.

In each of these three meta-analyses, for each article, the different concentrations of specific nanoparticles were differentiated, as well as cases in which the same concentration of nanoparticles was applied against different bacterial species. 

## 3. Results

### 3.1. Study Selection and Flow Diagram

In the initial electronic search, 194 studies were located in Pubmed, 220 in Scopus, and 210 in Embase, but none in the grey literature database. Out of the 624 publications identified, 245 were removed due to duplication, and 271 were excluded after the initial screening of titles and abstracts. Of the remaining 108 studies, the full texts were obtained and analyzed, discarding a further 26 articles, for the following reasons: studies focused on a cement used for a different purpose than dental bonding (six studies); publications not focused on evaluating properties corresponding to the objectives of the review (eight studies); studies that did not add nanoparticles to a dental material, and so only evaluated the antibacterial properties of the nanoparticles by themselves (five studies); studies that focused on a resin material but not as a dental bonding material (two studies); studies that did not correspond to the research question (four studies); and in vivo investigations (one study). Finally, 82 studies met all the inclusion criteria and were included in the qualitative synthesis. Only 11 of these studies could be included in the quantitative review. The PRISMA flow diagram (Figure 1) illustrates the entire selection process. 

### 3.2. Qualitative Analysis

Of the 82 studies selected for review, seven different types of dental restorative materials were identified. Figure 2A shows the distribution of the studies according to the type of dental material.

Of the 15 studies that considered a glass ionomer cement as the experimental material, six studied conventional glass ionomer cement [5,18,22,23,24,25], six studied glass ionomer cement modified with resin [7,11,26,27,28,29], two evaluated the incorporation of nanoparticles into both the latter materials [19,30], and another work evaluated the incorporation of nanoparticles into a glass ionomer cement used to cement orthodontic bands [31].

Regarding articles that evaluated dental adhesives and orthodontic adhesives, of the 19 studies that focused on dental adhesives, only three took the primer into account [32,33,34], ten investigated orthodontic adhesives, and one considered the modification of the primer alone [10]. It should be noted that only one study assessed the addition of nanoparticles to all the resin-based materials used in dental restoration (composite and dental adhesive) [35].

A wide variety of nanoparticles were investigated in the works reviewed but most assayed a single type. However, in some articles, other components were used in addition to the main nanoparticle type in order to increase their antibacterial properties. Other studies evaluated combinations of different nanoparticles. Figure 2B summarizes the different types of nanoparticles used in the studies included.

Eleven studies incorporated three different types of nanoparticle sequentially (nanoparticles of amorphous calcium phosphate, quaternary ammonium dimethacrylate, and silver), among which, two subgroups could be distinguished: studies that added the nanoparticles to a dental resin [36,37,38,39,40,41] and studies that added nanoparticles to a dental adhesive [32,33,42,43,44]. Both subgroups developed out of earlier studies published by teams of closely-related authors, who evaluated the progressive incorporation of the three types of nanoparticle in different concentrations, in order to obtain a dental material with optimal antibacterial activity.

Similarly, another group of studies evaluated the joint incorporation of other types of nanoparticles (nanoparticles of amorphous calcium phosphate, silver, 2-methacryloyloxyethyl phosphorylcholine, dimethylaminohexadecyl methacrylate, dimethylaminododecyl methacrylate); these could be divided into three subgroups. The first comprised four studies [26,27,28,29] that added nanoparticles to glass ionomer cement, analyzing the sequential addition at different concentrations of the group of outstanding nanoparticles, all using the same control group. Again, these studies were published by interrelated teams of authors. The second was made up of studies that integrated nanoparticles into resin-based materials [45,46,47,48,49,50]; all of them stressed that the nanoparticles were incorporated into a resin already containing amorphous calcium phosphate nanoparticles (NaCP nanoparticles). As in the previous subgroup, the teams of authors were interrelated, the only difference between the works being the type of monomer (dimethylaminohexane methacrylate (DMAHDM) or dimethylaminododecyl methacrylate (DMADDM)) incorporated into the resin. The last subgroup comprised studies that added the aforementioned nanoparticles to dental adhesives [34,51,52,53,54,55,56,57]. Again, the author groups were interrelated, but as in the previous subgroup, the monomer added to the resin varied from study to study, and furthermore, some studies incorporated nanoparticles into all the adhesive components, while others only added them to the primer.

Among the variety of nanoparticles incorporated in the different studies, the wide variations in the components and mechanisms used for their synthesis were remarkable. Of the 18 studies that incorporated silver nanoparticles (AgNP); 11 acquired commercially available nanoparticles [3,4,5,6,7,8,10,11,24,57,58], five synthesized them chemically from reagents [15,59,60,61,62], one [12] stated that AgNPs were chemically synthesized using the method described by Akhavan et al. (2014) [63], and one synthesized AgNP in situ from Ag Benzoate [31]. Among the studies that added TiO_2_ NP, four used commercially available nanoparticles [16,18,23,64], and one chemically synthesized them from commercially available TiO_2_ powder [9]. In studies that evaluated the addition of ZnO NP, four acquired them commercially [2,14,19,65], and one synthesized the nanoparticles using a CO_2_ laser [66]. Of the studies that incorporated NaCP NP, quaternary ammonium dimethacrylate (QADM), and AgNP, nanoparticle synthesis was used for all compounds: NaCP NP was chemically synthesized using a spray-drying technique, Ag nanopowder was commercially acquired, and QADM was chemically synthesized using a modified Menshutkin reaction. The same happened with studies in which NaCP NP, AgNP, 2-Methacryloyloxyethyl phosphorylcholine (MPC), DMAHDM, DMADDM, and 12-methacryloyloxydodecylpyridinium bromide (MDPB) were added: the silver nanopowder was commercially acquired, MDPB was commercially acquired, NaCP NP was chemically synthesized using a spray drying technique, MPC was commercially acquired, and DMAHDM and DMADDM were chemically synthesized using a modified Menshutkin reaction. Of those which incorporated quaternary ammonium polyethylenimine nanoparticles (QPEI NP), in seven works the nanoparticles were chemically synthesized using the method described by Beyth et al. 2006 [22,35,67,68,69,70,71], including the article of Beyth et al. 2006 that defined the synthesis mechanism [35], and the other two also mentioned chemical synthesis, without referring to Beyth et al. 2006 [72,73]. The three articles that added CuNP acquired them commercially. Of the remaining 13 articles, all of them referred to chemical synthesis from reagents, with the exception of Makvandi et al. 2015, who used commercially available nanoparticles [74].

The studies used different tests to assess antibacterial capacity. Most of the studies used a number of tests, with variability in the results. Figure 2C reports in detail the number of studies that use each of the cited antibacterial tests.

According to the inclusion criteria, each of the studies included a suitable control group that would make it possible to evaluate the effect of incorporating nanoparticles into the material on its antibacterial effect. In cases in which the nanoparticles were incorporated into a dental material synthesized from scratch, commercial material with a similar composition but without nanoparticles was used as a control group [8,36,37,38,39,40,41,42,45,46,47,48,49,50,53,54,75,76,77,78]. However, Argueta-Figueroa et al. used 0.2% chlorhexidine and different antibiotics as control [79].

Detailed information about the studies included for analysis is summarized in Appendix A: study groups (specifying the percentage of nanoparticle in each group and sample size); type of nanoparticle; type of dental bonding material (specifying the commercial name); storage and handling conditions; the antibacterial tests used; the groups of microorganisms evaluated and the culture broth used; conclusions; and quality.

### 3.3. Quality Assessment

As for the methodological quality of the studies, 43 studies [2,3,6,9,12,14,15,16,24,25,31,32,33,34,35,36,37,38,39,41,42,45,46,47,48,49,50,53,54,55,56,57,58,59,60,61,64,65,72,74,75,76,77,78,80,81,82,83] were classified as at high risk of bias, while the rest showed a medium risk of bias.

Generally, the items that obtained positive scores in most of the studies were: adequate control group, materials used according to the manufacturer’s instructions, standardized antibacterial assessment, and sample production process, and adequate statistical analysis. However, the item ‘calculation of sample size’ was only fulfilled in one study, and the items referring to the production of the samples by a single operator and evaluation of antibacterial properties by a single operator were not fulfilled in any. Figure 3 presents the distribution of the risk of bias according to the different items evaluated. Appendix A shows the score of each study for each of the items analyzed.

### 3.4. Quantitative Analysis

Of the 82 studies that met all the inclusion criteria and were included in qualitative analysis, only those that presented exact data, both at baseline and the end of the study period, in direct contact tests and colony-forming unit counts, were included in the quantitative analysis, 11 studies in total. Out of these studies, not all of them were included in each of the meta-analyses since different inclusion criteria were applied.

#### 3.4.1. Changes to Bacterial Activity in Orthodontic Bonding Materials 

All the studies obtained significant changes in bacterial activity, and meta-analysis estimated a Hedge’s G standardized mean difference of –4.91 (CI 95% from 5.99 to –3.83), with a Z-test p-value <0.001, indicating a considerable effect on bacterial activity (a decrease in the number of colonies in bonding materials modified by adding nanoparticles in comparison with control groups). Meta-analysis showed high heterogeneity, with a Q-valor of 143.9 (Q test p-value = 0) and an I-square value of 85.4 (Figure 4). Four different studies were included in the first meta-analysis [10,16,64,83].

#### 3.4.2. Changes to Bacterial Activity in Resin-Based Materials

Figure 5 shows the results of the second meta-analysis, comparing the antibacterial capacity of resin-based materials between materials incorporating nanoparticles and those without which includes three studies [3,6,58]. Of the ten groups included in the meta-analysis, only two did not obtain significant changes (Azarasina et al., 2013 a, AgNP 0.5% against *Streptococcus mutans*; Azarasina 2013 c, AgNP 0.5% against *Lactobacillus*). The addition of nanoparticles to resin-based materials significantly increased their antibacterial activity, showing a Hedge’s G standardized mean difference of –1′78 (CI 95% from –2.60 to –0.97), with statistically significant difference (Z-test p-value <0.001). Meta-analysis showed high heterogeneity, a Q value of 70.1 (Q test p-value = 0), an I-square value of 70.1 (Q test p-value = 0), and an I-square value of 87.15. 

#### 3.4.3. Changes to Bacterial Activity in all Dental Restorative Materials

When comparing overall variations in antibacterial properties of all the dental restorative materials with and without the addition of nanoparticles, a significant general effect was found due to the addition of nanoparticles (Z-test p-value <0.001), estimating a Hedge’s G standardized mean difference of –1.32 (CI 95% from –1.66 to –0.98), which indicated an important effect on antibacterial capacity. The meta-analysis showed heterogeneity, with a Q value of Q = 214.17, Q test p-value = 0, I-square = 79.7 (Figure 6). Ten different studies were included in the third meta-analysis [3,6,8,10,16,19,58,64,65,78,83].

When subgroups were analyzed in relation to the type of nanoparticles incorporated, no significant effect was found for the subgroup CaP (Hedges’s G= –0.65, CI 95% from –1.33 to 0.02, p-value = 0.058, test Z = –1.89) or the subgroup CaP/Ag (Hedges’s G = –0.44, CI 95% from –1.11 to 0.22, p-value = 0.191, Z test = –1.31). However, for the rest of the subgroups, significant effects on bacterial activity were found, whereby subgroups curcumin nanoparticles (CurcNP) (Hedges’s G = –1.84, CI 95% from –2.41 to –1.28, p-value = 0, Z test = –6.45) and TiO2 NP (Hedges’s G = –1.94, CI 95% from –2.88 to –1.01, p-value = 0, Z test = –4.07) produced the greatest significant improvements in antibacterial capacity (Figure 7).

On the basis of the first two meta-analyses focused on orthodontic bonding materials and resin-based materials, a meta-regression analysis was performed for each (Figure 8) to determine if the concentrations of nanoparticles incorporated in materials could vary the antibacterial effect. It was seen that nanoparticle concentration had no significant effect in either case, the R2 being 0 in both analyses.

### 3.5. Publication Bias

The risk of publication bias was assessed by means of funnel plots and fail-safe N values. The funnel plots (Figure 9) showed some asymmetry, and fail-safe N was estimated at 2215 for the meta-analysis of orthodontic bonding materials and 1997 for dental adhesive materials, which indicated that a large number of non-significant studies would be required for the estimation obtained in meta-analysis to cease to be significant.

## 4. Discussion

Incorporating nanoparticles into dental materials in order to improve the durability and efficacy of dental restorations is a new field of great interest. A wide variety of in vitro studies has focused on investigating different types of nanoparticles, their concentrations, and the optimal means of incorporating them into materials in order to obtain a dental material that is resistant to bacterial colonization. The present systematic review analyzed 82 articles, which were found to present high heterogeneity.

Although some literature reviews have focused on the incorporation of nanoparticles into dental materials, they have only evaluated the addition of nanoparticles to a specific material or studied a single type of nanoparticle. Elkassas et al. [84] summarized the current state of nanotechnology and provided some indications of the potential involvement of nanotechnology in preventive and adhesive dentistry. Allaker et al. [17] introduced the possibility of controlling oral infections through the use of nanoparticles, while Noronha et al. [85] examined the different antibacterial uses of silver nanoparticles in composites, explaining their mechanism of action and their most important toxicological aspects. But to date, no systematic review has conducted a quantitative analysis of the increase in antibacterial properties derived from adding nanoparticles to dental restorative materials.

All the studies analyzed in the present review were in vitro studies. In vivo studies were not considered due to the difficulties of standardization and comparison that these types of studies entail, which could lead to bias. In the case of Cheng et al. [33], these authors conducted both in vivo and in vitro trials, but only data from the in vitro assay were included in analysis. In vivo studies provide more reliable results, but in most cases, it is ethically and technically compromised to conduct them in the early periods of the investigation. In vitro studies are unable to simulate the exact oral conditions, which may lead to some biases, and the obtained results cannot be totally extrapolated to the clinic.

The fact that the studies included in the review used different control groups could lead to some biases and make comparisons between studies less accurate. However, these differences were justified since they used the same material they analyzed in their research but without the incorporation of nanoparticles, so comparisons within the same study were more accurate. 

Among the wide variety of nanoparticles used in the studies, silver nanoparticles were the most commonly assayed, either alone or in combination with others, such as zinc oxide nanoparticles [3], silver nanoparticle-laden hydroxyapatite nanowires [15], quaternary ammonium dimethacrylate [4], silica nanoparticles [8], or hydroxyapatite [12]. 

Regarding the type of nanoparticle, a possible drawback might be that most of the studies incorporate commercially acquired nanoparticles into the dental adhesion material, which might cause the incorporation of excipients or another type of material not considered in the composition.

In relation to the type of nanoparticle incorporated, each study recorded the nanoparticles’ mechanism of synthesis. It should be noted that only one of the studies considered the use of laser for nanoparticle synthesis [66]. Esteban-Tejada et al. [66] used a CO_2_ laser in their study, and although it was not used to generate the nanoparticles as such, it made it possible to assemble ZnO and CaO powders producing large quantities of nanometric and sub micrometric fibers. Although some studies have used lasers in the nanoparticle production process, to date, no published study has incorporated nanoparticles generated in-situ using laser into a dental material. At this point, many studies use lasers in their methods, but none of them has used these devices to strictly synthesize the nanoparticles. It would be interesting to analyze the results of studies that incorporate nanoparticles utterly synthesized by laser and compare the results with those studies that incorporate the same type of nanoparticle but commercially acquired.

Nanodiamonds (NDs) have made a recent appearance in the field of medicine and dentistry. Their potential is under investigation for use in different therapy platforms in various fields of medicine, such as drug delivery, tissue regeneration, and gene therapy. Recent research also suggests that NDs could also be used as bioactive or antibacterial dental implant coatings [86,87]. However, no study has been found that incorporates NDs into dental restorative materials and which meets the inclusion criteria of the present review [86,87].

The incorporation of nanoparticles into dental materials aims to produce materials that are more resistant to bacterial attack, and therefore to the appearance of secondary caries. *Streptococcus mutans* and *Lactobacillus acidophilus* are mainly responsible for the appearance of caries, and for this reason most of the studies reviewed used these bacteria in antibacterial tests. Even so, certain studies used other microorganisms in their antibacterial tests, such as *Escherichia coli* [25,59,66,67,74,79], *Staphylococcus aureus* [5,59,66,67,74,79], *Pseudomonas aeruginosa* [59,67,74], *Streptococcus sobrinus* [2,8,60], *Streptococcus sanguinis* [12,16,83], *Staphylococcus epidermidis* [67], *Streptococcus oralis* [2,66], *Streptococcus gordonii* [2], *Actinomyces naeslundii* [2,76], *Candida krusei* [66], *Porphyromonas gingivalis* [45,46,53], *Prevotella intermedia* [45,46,53], *Prevotella nigrescens* [45,53], *Aggregatibacter actinomycetemcomitans* [45,46,53], *Fusobacterium nucleatum* [45,46,53], *Enterococcus faecalis* [45,53,67,69,71], *Enterococcus faecium* [53], *Parvimonas micra* [53], *Lactobacillus casei* [22,69,76,81], *Actinomyces viscosus* [68,69], *Candida albicans* [74], and *Bacillus subtilis* [74].

In order to achieve more reliable results, it would be interesting to carry out bactericidal tests for each type of nanoparticle against different types of microorganisms. This way it would be possible to assess whether one type of nanoparticle has higher antibacterial activity against some specific microorganisms than against others, and also to compare the antibacterial capacity of the different nanoparticles against the same bacteria.

A large number of tests are available for evaluating the antibacterial capacity of dental materials, although five were the most commonly employed in the studies reviewed, the disc diffusion test and direct contact test being the most accurate and reproducible [14,61]. The disc diffusion test is one of the most frequently used methods, but it requires water-soluble components that can be released from the dental material’s composition. When the nanoparticles added are less soluble (zinc oxide, for example), this technique could not be used, and so the direct contact test should be used [65,88]. The direct contact test assesses the antibacterial efficacy of solid nanoparticles diluted in soluble substances. Kim et al. used the disc diffusion test to evaluate the antibacterial activity of a resin containing chlorhexidine nanoparticles, but as the chlorhexidine nanoparticles could not be released to diffuse through the culture medium, the authors extracted the chlorhexidine from the samples, using sterile paper discs soaked with the chlorhexidine for testing [89].

The studies could be grouped according to the type of material modified: resin-based materials, dental adhesives, and glass ionomer cement. While these materials were of multiple brands, most of the studies focusing on orthodontic adhesives assayed Transbond XT (3M), and only one study investigated a different brand: NeoBond (Dentsply).

Some of the studies did not specify the material’s manufacturer, either for internal reasons or because materials were especially synthesized for the study. This was one of the reasons that led to a decrease in the methodological quality of most of the studies.

Most of the studies included in the present systematic review agreed that the incorporation of nanoparticles into dental restorative materials leads to an increase in their antibacterial capacity when compared with the original unmodified material. Positive results were found for all types of dental materials included in the review; glass ionomer cement, resin-based materials, dental adhesives, orthodontic adhesives, temporary cement, resin-based cement, and resins used for sealing fissures. 

Nevertheless, a few studies did not reach this conclusion, failing to find significant relationships between the incorporation of nanoparticles and improvement in antibacterial capacity. Magalhães et al. concluded that the incorporation of AgNP into resin-based cement did not increase its antibacterial activity against *S. mutans* but entailed a noticeable color change and greater sorption in comparison with cement without AgNP [62]. The study by Garcia-Contreras et al. evaluated three types of material and only referred that one (FX-II Enhanced restoration) produced an increase in antibacterial activity as the result of adding TiO_2_ NP; however, the other two materials (core shade base cement and base cement), with or without modification, showed no antibacterial properties for any of the specimens [18]. Garcia et al. found that the inclusion of ZnO NP in glass ionomer cement did not promote antibacterial activity against *S. mutans* [19]. Despite the rise of nanotechnology and a large amount of research that is being carried out around this field, there is high variability among the different study groups, regarding the nanoparticle concentration to achieve the greatest antibacterial effect, as well as the required manipulation process. Perhaps the reason why these previous studies have not been able to show significant results lies in the use of very low concentration of nanoparticles, not being enough to show antibacterial activity, as well as the possible risk of nanoparticle agglomeration when the acidity of the medium is not considered.

Storage conditions and the manipulation of samples prior to antibacterial analysis was a factor that varied between studies and could be a cause of variation in the results. Not all the studies referred to a sample sterilization process before the antibacterial tests, and only 41 of the 82 studies specified the use of a specific protocol. Among these, a wide variety of techniques were employed, sterilization through ethylene oxide being the most commonly used [4,10,11,19,26,27,28,29,33,34,36,37,40,42,43,44,45,46,47,48,51,52,53,54,56,57,75,77]. Less commonly used sterilization techniques were: autoclave [3,58], gamma radiation [12,16,83], ultraviolet light [78,80,82,88,90], and incubation in 70% ethanol [30,91,92]. Eight studies did not describe the technique used for sterilization. 

Some studies described the preservation of samples in a liquid medium before carrying out antibacterial tests, in order to eliminate uncured monomers; however, the protocols differed widely between the studies using this method. El-Wassefy et al., Saffarpour et al., and Xie et al. immersed the samples in sterile water at 37 °C for 24 h [5,49,65], while others, such as Kasraei et al., stored the disks in distilled water at 24 °C for 24 h [3]. Some studies also used agitation techniques; Li et al., Melo et al., and Zhang et al. placed the samples in water and agitated them for 1 h [4,42,44], while others, such as Xie et al., Wang et al., and Zhang et al., stored the disks in 200 mL of distilled water and agitated them with a magnetic stirrer at a speed of 100 rpm for 1 h [27,45,54]. Some studies combined both techniques: Chen et al. agitated the disks in water for 1 h and then immersed them in distilled water at 37 °C for 1 day [51].

Only eight of the 82 studies polished the samples before the antibacterial tests. Four of them [3,81,82,88] used the same technique, polishing samples with silicon carbide paper. The four other works employed different techniques. Aydin Sevinç et al. used Sof-Lex discs [2], das Neves et al. polished the samples with Enhance drills [6], Natale et al. stated that the samples were wet-polished [78], and Pietrokovski et al. employed polishing diamond rotary instruments [68]. Polishing protocols before the antibacterial analysis could lead to differences between the studies. Although polishing might be considered a better method to reduce bacterial activity, smooth surfaces are less retentive, so adhesion might be affected.

The incubation period of samples was found to be very heterogeneous, even though the results of antibacterial tests were dependent on this parameter. Incubation intervals varied between 1 min in some studies [62] and 30 days in others [61]. Very short incubation periods are often related to the absence of bacterial inhibition due to insufficient action time. In fact, studies that failed to find an improvement in an antibacterial activity used short incubation periods. Magalhães et al. [62] applied four different incubation periods: 1 min, 5 min, 1 h, 6 h, and 24 h, while Garcia et al. [19] and Garcia-Contreras et al. [18] incubated the samples for 24 h. Although different sample incubation protocols make comparisons less accurate, the differences in the protocol between the different studies might be due to the fact that each type of bacteria needs different incubation periods.

The risk of bias among the studies included in the review was high in 43 cases, and medium in the remaining 39. The potential risk of bias was mainly due to technical shortfalls for calculating sample sizes and/or for handling or preparing samples. Only one study [30] specified the technique used to calculate sample size; all the other works either failed to apply any technique or failed to describe it. The handling of materials constitutes a factor that could influence antibacterial properties, whereby, unless handled correctly, the material might not produce the optimal effect. In this context, only half the studies stated that the materials had been used according to the manufacturer’s instructions. But it should be noted that in nine works, the negative point awarded in the scoring risk of bias was questionable, as they did not use branded materials but synthesized them from scratch, and so they were unable to follow an established protocol supplied by the manufacturer [45,46,47,48,49,50,53,54,78].

Another cause of medium to high risk of bias was the failure to stipulate whether single or multiple operators had participated in sample production and antibacterial testing; none of the studies stated that a single operator had carried out these processes.

The quality of the studies was evaluated taking as reference what the authors specified in the study document, perhaps some of them fulfilled positively more items than the reported ones, but as they did not specify it as such in the document, they were marked negatively, obtaining a high risk of bias. As a consequence, the medium-high risk of the studies included in the qualitative analysis led to low evidence of results.

The quantitative analysis performed in the present review included a total of 11 works, only those that assessed antibacterial activity by means of the direct contact test, as this test is more reproducible and produces comparable results [61]. Despite the fact that some works evaluated variations in antibacterial capacity over periods of up to thirty days, to avoid bias and standardize groups, only data obtained during the first 0–72 h incubation were used for analysis, given that this was the period that most of the studies focused on. For those studies that included samples with different concentrations of nanoparticles, analysis differentiated between each value obtained, as well as values obtained when the same concentration was applied to different bacterial species. Quantitative analysis was performed by standardizing the mean values obtained, but it should be noted that the units used across different studies differed and were sometimes impossible to standardize.

The first meta-analysis only included studies of different nanoparticles added to orthodontic bond materials, evaluating their antibacterial properties. The results of meta-analysis pointed to a positive effect on antibacterial capacity resulting from the incorporation of nanoparticles, with greater effects found in some studies than others (Figure 4); however, it should be remarked that most studies presented a high risk of bias. The most significant effects were obtained by Sodagar et al. for 5% and 10% concentrations of TiO_2_ on *S. mutans* and *S. sanguinis* [16]. Although these results pointed to greater antibacterial capacity for high concentrations of TiO_2_ against *S. mutans* and *S. sanguinis*, the effect obtained was far less against *L. acidophilus*. But the study by Sodagar et al. [83] did not follow the same pattern; samples showed less antibacterial potential than Sodagar et al. [16], while samples modified with CurcNP showed better results against *L. acidophilus*, regardless of the CurcNP concentration [83]. These findings could mean that TiO_2_ nanoparticles are more effective against *S. mutans* and *S. sanguinis* colonies, and CurcNP nanoparticles are more effective against *L. acidophilus*, although further research with a more clearly defined protocol is required to confirm this hypothesis. 

The second meta-analysis included studies that incorporated nanoparticles into resin-based materials, evaluating the antibacterial effect on this specific type of material. It was seen that the addition of nanoparticles had a significant effect on their antibacterial activity (Figure 5); however, higher-quality studies would be needed to obtain more significant results.

Although most of the data analyzed reflected a positive effect on the resin-based materials assayed, two works obtained notable results, showing the most significant changes: Kasraei et al., who tested 1% Ag nanoparticles against *S. mutans* and 1% ZnO nanoparticles against *S. mutans* [3], obtaining results that could be the source of the heterogeneity observed in this meta-analysis. The results showed that Ag and ZnO nanoparticles showed a much greater antibacterial effect on *S. mutans* than on *Lactobacillus* [3].

Many of the studies included in the qualitative analysis did not provide exact data so that they could not be included in quantitative analysis, which greatly limited the number of works in the first two meta-analyses; this might have decreased the precision of the comparisons made.

The third meta-analysis included all the materials used for dental restorative procedures, modified with different types of the nanoparticle. This allowed a broader and more detailed analysis of the effect of nanoparticle incorporation on antibacterial activity, making it possible to make comparisons between nanoparticle types. This meta-analysis also found a significant effect on the materials’ antibacterial capacity produced by the addition of nanoparticles, with more significant effects in some groups than others, which could explain the heterogeneity of the analysis: Poosti 2013 assayed 1% TiO_2_ against *S. mutans* [64]; Kasraei 2014 1% Ag against *S. mutans* [3]; Kasraei 2014 1% ZnO against *S. mutans* [3]; Degrazia 2016 0.11% Ag against *S. mutans* [10]; Degrazia 2016 0.18% Ag against *S. mutans* [10]; Degrazia 2016 0.33% Ag against *S. mutans* [10]. In particular, data obtained by Poosti et al. showed that TiO_2_ was found to have a potent antibacterial effect on *S. mutans* [64], while Degrazia et al. observed a powerful antibacterial effect of Ag against *S. mutans* even at low concentrations [10]. The overall results of this meta-analysis showed that the incorporation of nanoparticles into restorative materials led to an increase in antibacterial activity. However, eight of the included studies showed the high risk of bias due to failure to report items, such as sample size calculation, material manipulation following the manufacturer’s instructions or conducting the experiments by a single operator, which points at the need of further studies with more accurate designs or better reporting of the methodology.

Some studies using the same type of nanoparticle obtained different results [3,8,10]. This could be attributed to the variations in the nanoparticle concentrations. However, the meta-regression conducted in the present study showed that the nanoparticle concentration did not affect the antibacterial results significantly. Since these three studies used the same microbial species (*S. mutans*), the difference could not be attributed to this factor either. Therefore, differences in results might be owed to differences in other methodological aspects. The studies conducted by Poosti et al. and Sodagar et al. also obtained different results when using the same nanoparticle type. In this case, the variation could be attributed to the use of different bacterial species (*S. mutans* and *L. acidophilus*, respectively) [16,64].

The three meta-analyses showed high heterogeneity and a significant effect derived from the incorporation of nanoparticles into dental restorative materials. It might be stated that, although further research is required with more specific objectives to generate stronger evidence, the incorporation of nanoparticles into dental restorative materials, regardless of their composition, produces a considerable increase in their antibacterial activity. Moreover, contrary to what might have been expected, the meta-regression analysis performed in the present study revealed a significant relation between nanoparticle concentration and antibacterial effect.

Since the results of the present review were based on in-vitro studies, it should be mentioned that nanoparticles might play a secondary role in in-vivo conditions. In the oral environment, other protective factors, such as mechanical plaque control, diet control, or the application of pH neutralizing techniques, might be involved [93]. Furthermore, since acid production is the main cause of caries, which is influenced by both the phenotypic and genotypic properties of the oral microflora, oral conditions should always be considered [94].

Bearing in mind that many environmental factors depend on the patient’s cooperation, whenever patients do not cooperate, the antimicrobial effects of the restorative materials gain importance. Additionally, although WSL is mostly reversible [13], it is avoiding their emergence using restorative materials with antibacterial effect since their treatment relies on patient’s cooperation and also when these lesions are covered by restorations could go unnoticed so become irreversible.

The technique of reinforcing dental materials with nanoparticles is a recent development. Although the earliest reports were published in the period 2006–2008 (Beyth et al. [35], Beyth et al. [67], Yudovin-Farber et al. [72]), the main body of literature has been published in the last five years. Most studies investigating this field have reported beneficial results, which points to the scope for improving the antibacterial properties of dental restorative materials by incorporating external nanomaterials. While this line of research has produced promising results so far, the high heterogeneity of the three meta-analyses conducted in the present systematic review, together with the high risk of bias found in most of the studies analyzed, point to the need for further studies with standardized protocols.

## 5. Conclusions

The qualitative analysis of the included studies shows that although there is a large background of research around nanotechnology, the criteria regarding the nanoparticles concentration, culture broth and incubation period, storage conditions, and manipulation of samples vary considerably between the different studies.

Despite the heterogeneity of results identified in the present meta-analyses, it might be concluded that the incorporation of nanoparticles into dental bonding materials is a positive means of increasing their antibacterial properties.

The antibacterial activity of nanoparticle-modified dental bonding materials is significantly higher when compared with the original materials.

TiO_2_ nanoparticles offer the greatest antibacterial benefits when incorporated into a dental bonding material.

The heterogeneity observed across the studies reviewed highlights the need for further research and the design of standardized study protocols in terms of nanoparticle type, concentration, and incorporation method.

## Figures and Tables

**Figure 1 medicina-56-00055-f001:**
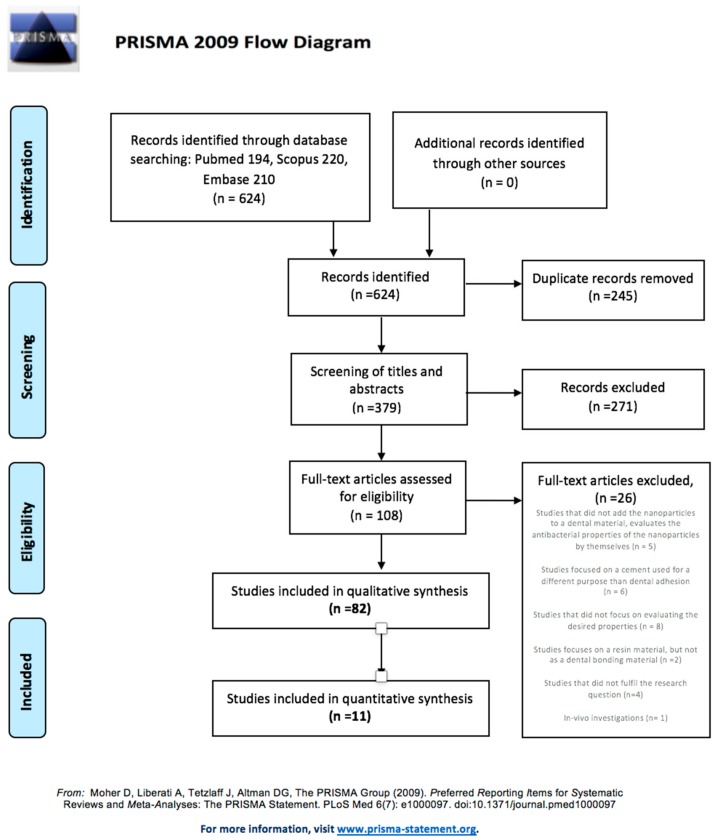
The PRISMA flow diagram. From: Moher D, Liberati A, Tetzlaff J, Altman DG, The PRISMA Group (2009). Preferred Reporting Items for Systematic Reviews and Meta-Analyses: The PRISMA Statement.

**Figure 2 medicina-56-00055-f002:**
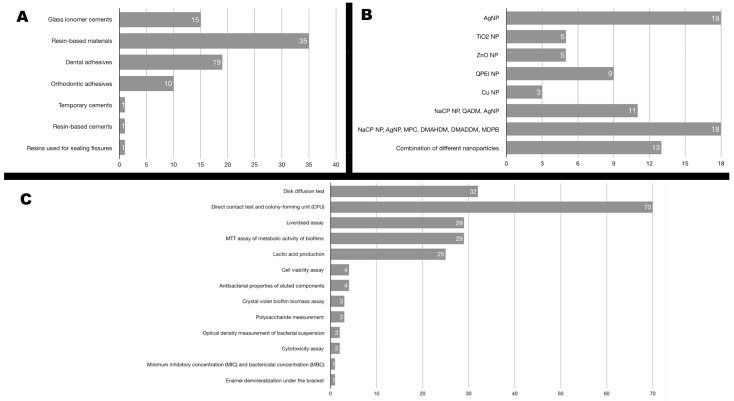
Distribution of the studies according to (**A**) the type of dental material, (**B**) the type of nanoparticle incorporated, (**C**) the type of antibacterial test used to assess antibacterial capacity.

**Figure 3 medicina-56-00055-f003:**
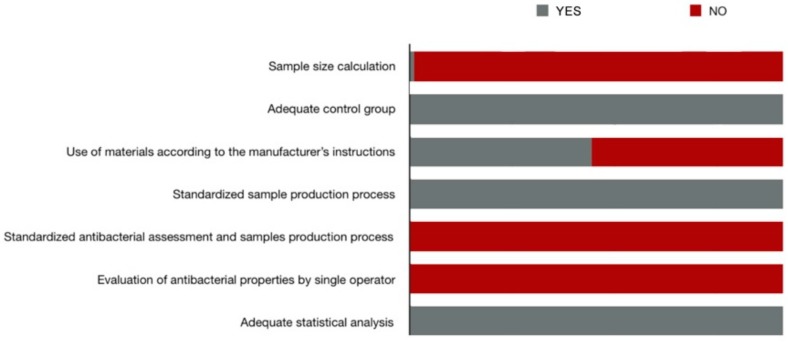
Risk of bias diagram.

**Figure 4 medicina-56-00055-f004:**
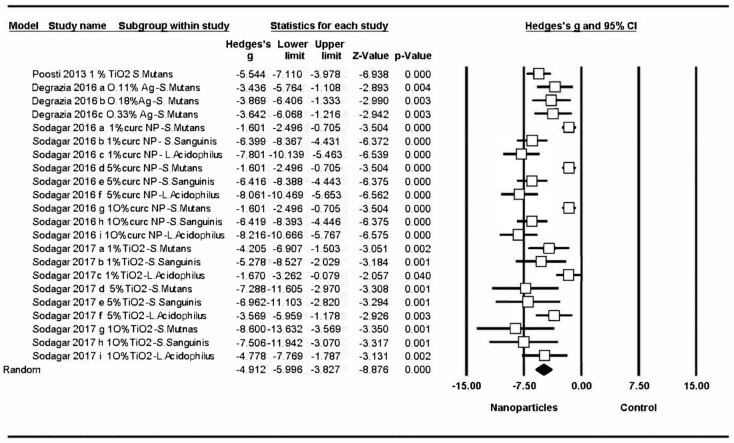
Forest plot summarizing the antibacterial activity of orthodontic adhesives.

**Figure 5 medicina-56-00055-f005:**
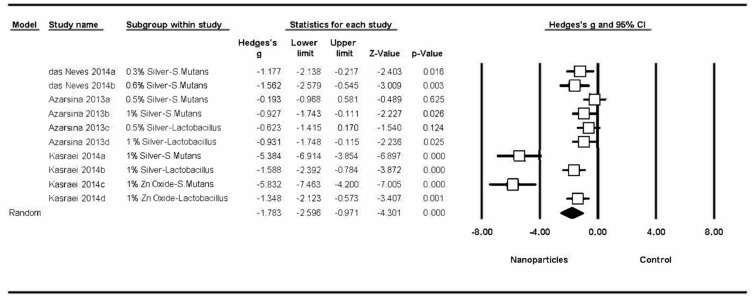
Forest plot summarizing the antibacterial activity of resin-based materials.

**Figure 6 medicina-56-00055-f006:**
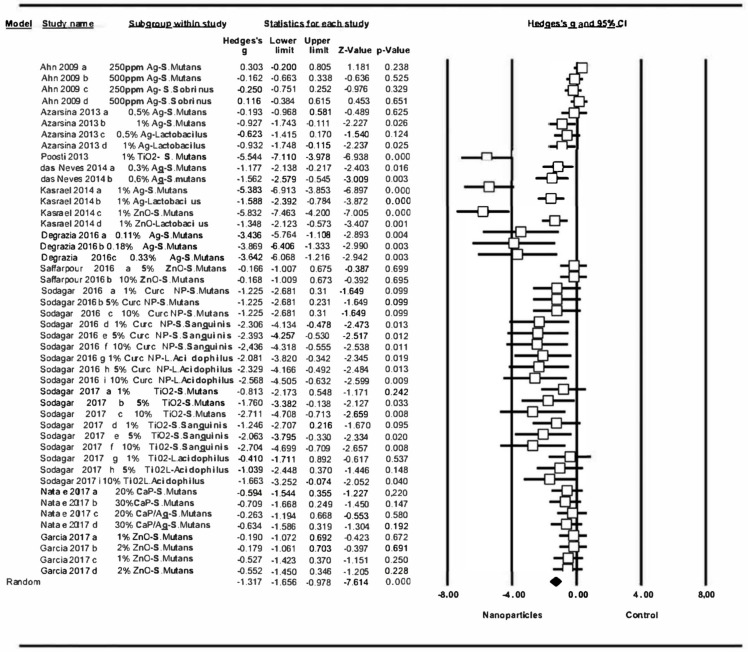
Forest plot summarizing dental restorative materials antibacterial activity.

**Figure 7 medicina-56-00055-f007:**
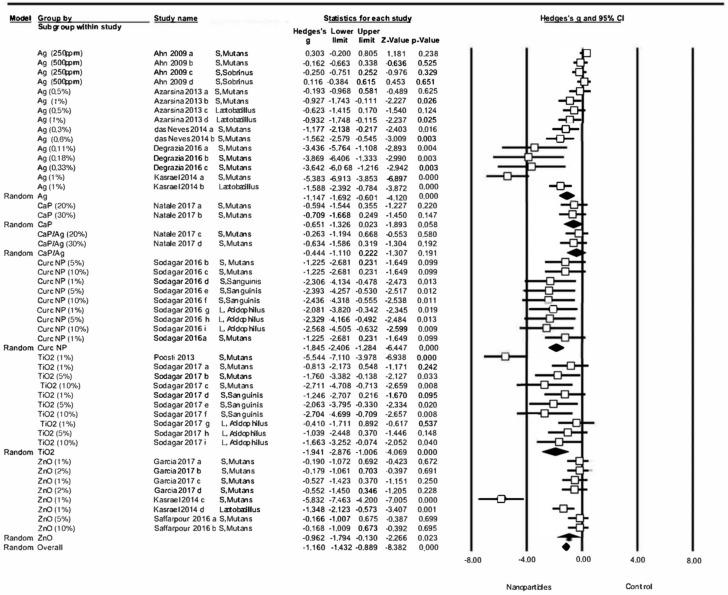
Forest plot summarizing the antibacterial activity of dental restorative materials organized by nanoparticle type.

**Figure 8 medicina-56-00055-f008:**
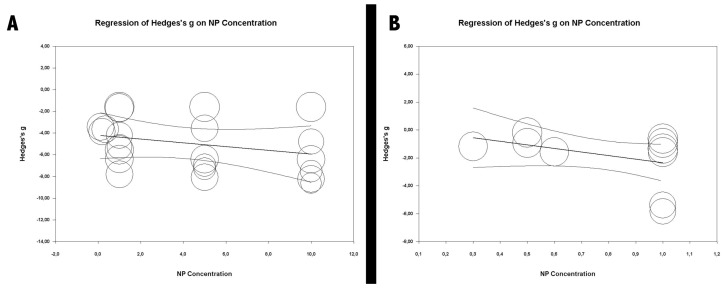
Meta-regressions according to nanoparticle concentration: (**A**) orthodontic adhesives’ meta-regression according to nanoparticle concentration; (**B**) resin materials meta-regression according to nanoparticle concentration.

**Figure 9 medicina-56-00055-f009:**
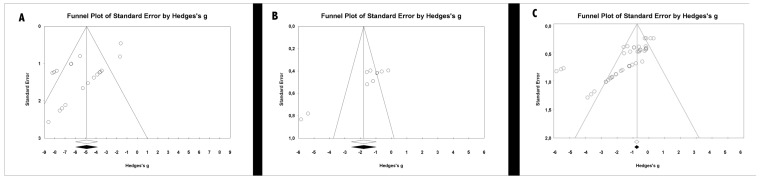
Funnel plots: (**A**) funnel plot of the antibacterial activity of orthodontic adhesives; (**B**) funnel plot of the antibacterial activity of resin-based materials; (**C**) funnel plot of the antibacterial activity of dental bonding materials.

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
