# Peer review of "Antibacterial Properties of Nanoparticles in Dental Restorative Materials. A Systematic Review and Meta-Analysis"

_medicina, 2020, doi:10.3390/medicina56020055_

Round 1

Reviewer 1 Report

The article is well written and demands minor changes. The aim is properly stated. The discussion is well written. Conclusions are based on results and answer the aim.

Introduction

It should focus more on the research topic. It can be shortened.

Figure 1 please correct the description – delete ‘?’. Now it is illegible.

Qualitative analysis

Some of the results could be presented as a table.

Reviewer 2 Report

This review provides qualitative analysis and quantitative analysis, is well written. Here are some suggestions.

redraw figure 1. add citations for line 443-445. Most of your conclusions are drawn from quantitative analysis, which comes from only 11 references. In qualitative analysis, you need to provide a deeper discussion and find the gaps, instead of solely categorize the references.

Reviewer 3 Report

#1 The authors claimed that they were evaluation “dental bonding materials”. In Figure 2, of the 7 material types listed, only one is regarded as traditionally accepted dental bonding material. The other are luting cements and restorative materials. In some cases, glass ionomer cements may be regarded as dental bonding material. Please change title and scope for the evaluation, or include only the studies with materials regarded as dental bonding materials

#2 The stated research question (Does the incorporation of nanoparticles into dental bonding materials increase their antibacterial/antimicrobial properties? does not concur with that stated in the Abstract. Please correct.

#3 Inclusion criteria:

Please delete first bullet point (In vitro studies) as it is included in third bullet point.

#4 Data extraction:

#5 How was the “quality of the article” assessed, or did the authors mean quality of the study?

#6 The authors stated that 11 studies were included in “Quantitative synthesis”. Are these studies listed in Figures 4-7? Please clarify.

#7 The data given in the Figure legend to Figures 4-7 must be incorporated in the figures for clarity.

#8 Noticed that there was no variation in control values in Figures 4-7. Please comment.

#9 43 studies were classified as at high risk of bias, however, these studies need to be identified and their possible influence taken into consideration in the results and discussion. Please clarify.

10 According to Figure 4, none of the evaluated studied had standardized antibacterial assessment and sample production process. Since antibacterial effect must be regarded as the most important endpoint, this was surprising and discouraging. Please comment.

#11 Part of the discussion (lines 386-391 and 401-422) is results and should be moved to that section.

Author Response

Please see the attachment (Response to Reviewer 3)

Round 2

Reviewer 2 Report

After your revision, this paper now is quite well written.

Author Response

No changes requested by the reviewer

Reviewer 3 Report

#1 In M&M, line 86, it is stated as inclusion criteria:

Randomized controlled trials, case-control studies, in vitro studies, and cohort studies in humans.

In Discussion, line 381: In vivo studies were not considered, due to the difficulties of standardization and comparison that this type of studies entails, which could lead to bias.

Please correct inclusion criteria or explain why include in vivo studies and the exclude them.

#2 Figure 5

A study of deNeves et al 2014 [ref 6 is included. This study in not listed among the quantitative studies in Figure 6. Was the number of quantitative studies actually 11? Please clarify.

#3 Risk of bias. The author have a general statement about risk of bias: line 532: “As a consequence, the medium-high risk of the studies included in the qualitative analysis leads to low evidence of results.” This is fine, however, when discussion further the results this seemed to be forgotten, please see # 4-6 below:

#4 Discussion, line 546 ff

The discussion of the results in Figure 4 does not take into consideration that 3 of the 4 studies were rated as having high risk of bias. Please modify.

#5 Discussion, line 559 ff

The discussion of the results in Figure 5 does not take into consideration that  all 3 studies were rated as having high risk of bias.  The authors even stated “It was seen that the addition of nanoparticles had a significant effect on their antibacterial activity (Figure 5)” . Please modify.

#6 Discussion. Line 571 ff

The discussion of the results in Figure 6 does not take into consideration that 7 of the 10 studies were rated as having high risk of bias. Even here the authors state “This meta-analysis also found a significant effect on the materials’ antibacterial capacity ……” Please modify.

#7 Discussion, general

The various concentrations used of NP in different studies seems not be elaborated upon. For examples, the effect of AgNP in the study of Ahn et al 2009 was much less than that of Degrazia et al 2016. Is there an explanation for that?
